# Model Stitching: Looking For Functional Similarity Between Representations

**Adriano Hernandez, Rumen Dangovski, Peter Y. Lu & Marin Soljacic**
MIT EECS
Cambridge, MA 02139, USA
`{adrianoh,rumenrd,lup,soljacic}@mit.edu`

## Abstract

*Model stitching* (Lenc & Vedaldi 2015) is a compelling methodology to compare different neural network representations, because it allows us to measure to what degree they may be interchanged. We expand on a previous work from Bansal, Nakkiran & Barak which used model stitching to compare representations of the same shapes learned by differently seeded and/or trained neural networks of the same architecture. Our contribution enables us to compare the representations learned by layers with different shapes from neural networks with different architectures. We subsequently reveal unexpected behavior of model stitching. Namely, we find that stitching, based on convolutions, for small ResNets, can reach high accuracy if those layers come later in the first (sender) network than in the second (receiver), *even if those layers are far apart*.

This leads us to hypothesize that stitches are not in fact learning to match the representations expected by receiver layers, but instead finding different representations which nonetheless yield similar results. Thus, we suggest that model stitching, naively implemented, may not necessarily always be an accurate measure of similarity.

## 1 Introduction

The success of deep learning for visual recognition has been attributed to the ability of neural networks to learn good representations of their training data Rumelhart et al. (1986). That is, intermediate outputs (which we refer to as "representations") of good neural networks are believed to encode meaningful information about their inputs, which these neural networks use for classification and/or other downstream machine learning tasks Goodfellow et al. (2016). However, our understanding of these representations is somewhat limited. Though deep learning interpretability research, particularly for computer vision, has helped us to intuitively grasp what deep neural networks are learning, we do not know why good representations are learned, nor do we have a robust theory to characterize them. For example, we do not know how to compare representations effectively.

Our goal is to improve the existing toolbox to find *functional* similarity between representations. It is not obvious how to find functional similarity, nor is it obvious exactly what it precisely means even though we (the authors) have some intuition of it, so before we continue we provide a sufficient definition for our purposes. For us, a representational similarity metric is good at measuring "functional similarity" if we can easily use it to tell whether two representations of one or two models are used by the model(s) for the same or similar purpose(s). We believe this is a useful lens because if two representations can be used for similar purposes then in some sense they encode similar information. We care about similar information because understanding whether two models's representations encode similar information could be useful for soft guarantees of safe, fair, or robust AI.

34th Conference on Neural Information Processing Systems (NeurIPS 2020), Vancouver, Canada.

While we are interested in functional similarity, many papers Kornblith et al. (2019) Morcos et al. (2018) Ding et al. (2021) look for measures of statistical or geometric similarity[1] because they can confirm known edge cases, like a pair of identical representations or a representation and a vector of random noise.

These measures are a great starting point, but are not very informative in general, since different neural networks could process and store information in ways which are analogous to eachother but not numerically similar. In fact, they have been found Ding et al. (2021) Dujmović et al. (2022) to be misleading on occasion in both Computer Vision and Brain Sciences.

We believe that a previous work Bansal et al. (2021) provides us with a potentially better functional similarity measure. It uses, learned transformations to *translate* representations from one layer into those for another layer. Their technique measures functional similarity, because invariant to the type of transformations used, it tests whether two representations can be interchanged, which is a strong indicator that the two representations function similarly. However, their work can only compare representations with the same shapes. We expand it to include all representations taking the form of ResNet tensors with widths and heights that are multiples of each other.

## 2 Experimental Setup

### 2.1 Models and Dataset

We compare all different layers of all ResNets with a number of layers ranging from ten to eighteen. These ResNets are trained on CIFAR-10 for comparable results with Bansal et. al. These small ResNets we characterize with 4-tuples, where each element is either one or two, representing the number of residual blocks per stage[2]. Since we use at most two blocks per stage, we can denote these 4-tuples unambiguously as $R_{1111}$, $R_{1112}$, and so on. There are $2^4 = 16$ such ResNets. Note that $R_{2222}$ is equivalent to the well-known Resnet18 architecture, while $R_{1111}$ is equivalent to Resnet10.

### 2.2 Experiment

We train each possible Small Resnet on CIFAR-10, yielding an accuracy above 90%. We also generate a randomly initialized, untrained network for each Small Resnet architecture and confirm that these have an accuracy of around 10%[3]. All these networks are frozen and cannot learn during stitching.

We stitch every ordered pair of Small Resnets. There are $16 \cdot 16 = 256$ such pairs. In every ordered pair of networks being stitched, the former is called the *sender*, and the latter is the *receiver*. A stitch is used to transform the output of the sender at an intermediate layer before inputing it into an intermediate layer in the reciever. For any network $A$ we can consider layer $i$ as $A_i$, the first $i$ layers (assuming we start at zero) as $A_{<i}$, and the layers after $i$ as $A_{i<}$. If we wish to include layer $i$ we can always call such (partial) networks $A_{\leq i}$ or $A_{i\leq}$. For an input $x$, if we call the sender $A$, the reciever $B$, and the stitch $S$, we call $C = B_{j<}(S(A_{\leq i}(x)))$ the *stitched network*. Normally, we train $S$ by doing backpropagation on the stitched network with both $A$ and $B$ frozen. The resulting accuracy is used to find the similarity[4] between $A_{\leq i}(x)$ and $B_{\leq j}(x)$, the former of which is called the *provided representation* and the latter of which is called the *expected representation*.

Unlike Bansal et al., which only compare corresponding blocks (i.e. $i = j$) of a sender and reciever with the same architecture, we stitch from *all* residual blocks of the sender into *all* residual blocks of the receiver even when they have different architectures, as long as they are Small Resnets. Also unlike Bansal et al., we only vary our neural networks by their initialization weights, but our setup is otherwise nearly identical. To be able to stitch between all blocks, we use strided convolutions or upsampling when the dimensions are not the same. In our case the heights and widths vary by

---

[1]Here, we mean simply that the representations, given proper shifts and rescales, are numerically or geometrically close by on average.

[2]Residual blocks are partitioned into four stages of consecutive blocks, within which they have the same shape. At each stage, the width and height halve, while the width doubles He et al. (2016).

[3]There are ten classes in CIFAR-10.

[4]We use the downstream accuracy of the stitched network as our similarity measure for simplicity, since the original networks' accuracies were over 90%. Otherwise, a ratio or difference, as used in Bansal et. al. may be more informative. In our case, no choice amongst these would change our qualitative result.

powers of two so we can simple use 2x2, 4x4, or similarly-sized convolutions to downsample. For upsampling we use 2x, 4x, or similarly sized 2D nearest-upsampling (meaning that an element is copied into a grid of equally-valued elements) before applying a 1x1 convolution in the stitch.

We use the randomly-initialized ResNets as controls. Our controls enable us to make sure that the stitches are appropriately powerful. By powerful we mean how complex the functions are that stitches can represent. If a stitch is very powerful, then even with random networks it should be able to yield high downstream accuracy because it can learn any transformation. In this case our stitches would always yield high "similarity" and therefore be uninformative. We can be sure this is not the case by ensuring that the stitches never yield high similarity for random networks (where only overly powerful stitches would).

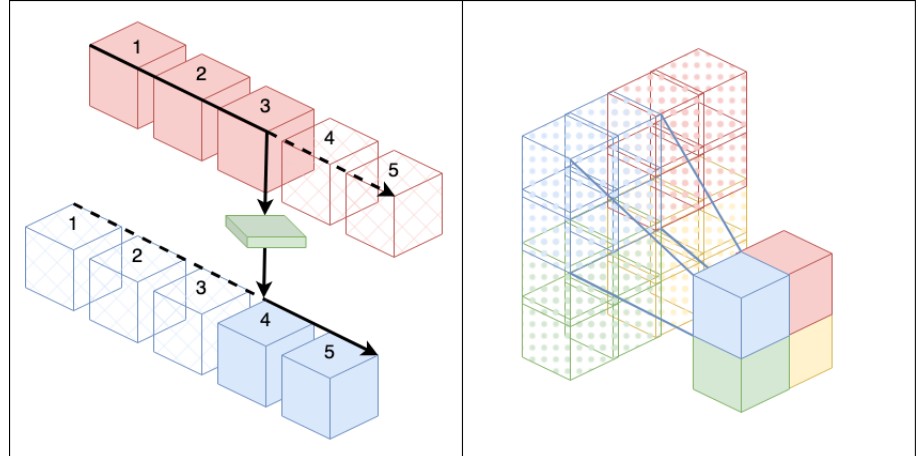

Figure 1: **On the left**, a diagram exemplifying a stitch from the red (sender) network into the blue (reciever) network comparing layer 3 from both. In this diagram, the blue layer 3 is the expected representation. Unused layers in the stitched network are displayed as partially translucent. The stitch is depicted in green. The arrows denote the flow of computation. The dashed arrows denote the flow of computation in regular operation, absent of stitching. **On the right**, a diagram exemplifying a 2x2 convolution (downsampling) from left (dotted) to right (solid). A 2x2 convolution such as this one could be used to stitch from a representation with larger width and height to one of smaller dimensions. The colors are chosen so as to elucidate which elements correspond. The blue lines further highlight the correspondence for the blue elements. In the case of upsampling, the image can be read from right (solid) to left (dotted), where the solid blue element is copied four times to the dotted blue elements before it is used (later) for a 1x1 convolution.

## 3 Results

For every ordered pair of networks, we plot the accuracy of all the stitched networks on a grid, based on the sender's layer and the receiver's layer. The layer is denoted by an integer which counts how many residual blocks came before it[5]. The value in the grid element is the accuracy of the stitched network after traning. Thus, this grid is a *similarity matrix* where entry $i, j$ corresponds to the similarity of $A_{\leq i}(x)$ and $B_{\leq j}$.

### 3.1 Expectations

We hypothesized that for similarity matrices between networks of the same architecture we would see a high similarity diagonal. For networks of different architectures we hoped for a shorter diagonal (since the matrix isn't square) or a diagonal with a different slope.

---

[5]The initial convolution is "0," the first block of the first stage is "1," and so on.

Generally, however, we assumed that there would exist some non-negligible number $\epsilon$ such that if we mapped each sender layer's representation to its most similar counterpart in the reciever, and their similarity was $s$, the similarity between that layer's representation in the sender and every other layer's in the reciever would be less than $s - \epsilon$. Moreover, we assumed that such a mapping would be injective. Intuitively, we thought that there would be a one-to-one correspondence between most layers in the sender to most layers in the reciever. We did not expect any layers' representation in the sender to have high similarity to *multiple* layers' representations in the reciever.

These hypotheses make sense because Bansal et. al.'s findings suggest that each layer in the sender should have at least one similar layer in the reciever—at least in the cast of identical architectures, where those two layers are the corresponding ones; and it is usually assumed that distant layers have different information, and so they should not be similar. However, **we found that every layer in the sender was extremely similar to all layers before it in the reciever by proportion of neural network length**. That is to say, for small Resnets, regardless of the architecture, if we had one Resnet of length $I$ and another of length $J$, then if $\frac{j}{J} \leq \frac{i}{I}$ the similarity was high between layer $i$ in the $I$-length network and layer and $j$ in the $J$-length network. Visually, this looks like a triangle in the lower left-hand corner of the similarity matrix. The triangle's endoints are the top left cell, the bottom left cell, and the bottom right cell. This pattern is visible in the figure below and quite perplexing.

For our controls we expected to see low stitching accuracy throughout the board since the networks are random, and we did. With that said, some of the top left or bottom right elements have high similarity depending on whether the sender or reciever was random. In the case of a random sender and trained reciever, if $i$ and $j$ are small, it is easy for $S$ to undo $A_{\leq i}$, and give $B_{j<}$ something usable. The opposite case is analogous.

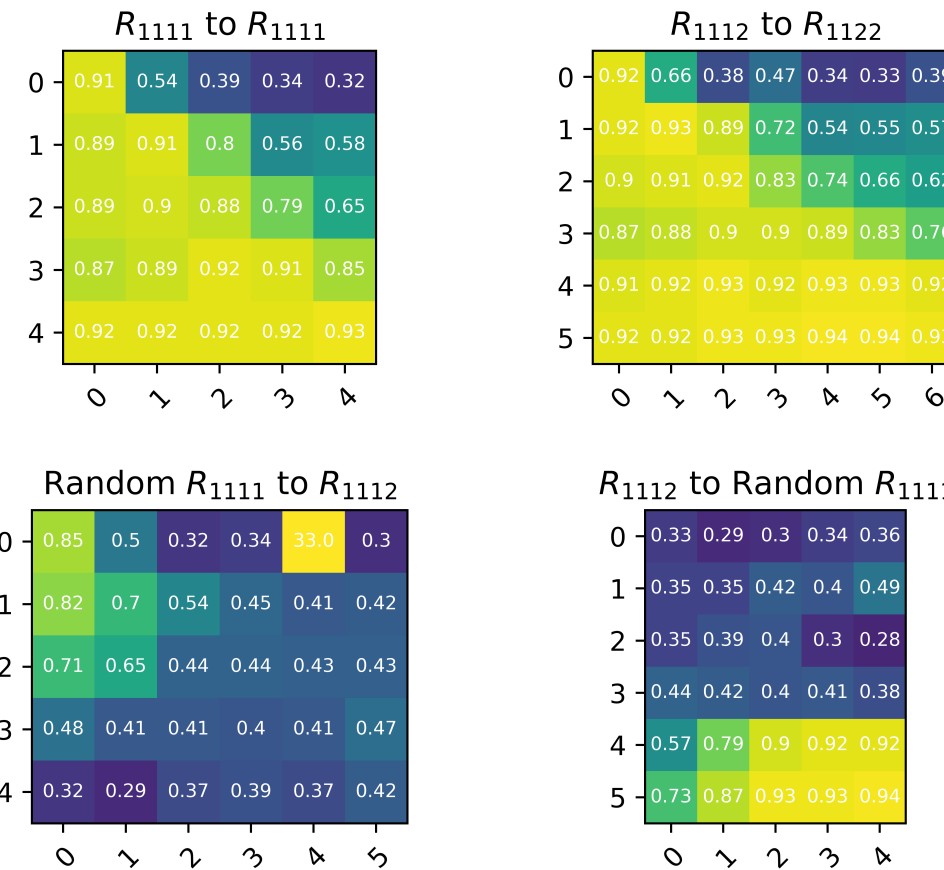

Figure 2: Triangle similarity pattern between trained Small Resnets and (expected) low similarity pattern for random ones. The plot is to be interpreted as a similarity matrix from sender to reciever. This is indicative of the pattern we saw on all such Resnets.

## 3.2 Conclusion

The most interesting aspect of our results is the high accuracy of the stitching network for layers in the lower left hand triangle. It is true that our findings do *not* contradict Bansal et. al.'s findings since they *only stitched on the diagonal* which yielded the same high accuracy for them as it did for us. However, we are still surprised. Given that we interpret the stitching network accuracy as a similarity, our results suggest that each sender representation is similar with *all* the receiver representations from a layer (proportionally) before it. We expected to see that each layer would be similar to a couple (nearby) layers at most because the standard narrative has been that every layer loses some amount of granular information, and so that information should not be reconstructable in an *interchangeability* test like stitching.

In our conclusions we focus our scope on the fact that layers are similar to those before them for similar, but not equal, length networks, instead of the fact that those layers "before them" need only be proportionally (to their own network's length) before, since most of our networks did not vary in length very much. However, the poportionality finding could prove to be interesting to explore in future work, since it may tell us something about Resnet length-invariant properties. A slightly more extreme case for the curious in depicted in the Appendix.

We see two main explanations for our results. The first is that the common narrative could be wrong and some neural networks may in fact be able to maintain most if not all of the granular information of the image throughout their processing of it. An alternative, but not uncommon, narrive to the standard one is that neural networks progressively discard information from layer to layer until only the class information remains. In that view, our results are actually expected since the stitch need only make sure that it transfers the class information. This could make sense since nearest upsampling is lossless, while downsampling is lossy, which matches the observation that stitching backwards yields high accuracy but not forwards, suggesting that it is easier to retain the information in the backwards direction.

The second explanation we see is that the stitch may be able to give the reciever a representation which, despite being different from that which is expected, nonetheless yields high accuracy. Intuitively, the stitch may be able to figure out how to generate some generic, albeit unrealistic, set of the most salient features for the recieving layer to classify in a given way. Perhaps it is generating the "average" human, or some sort of adversarial representation.

To rigorously determine why our obervations are as they are, however, a deeper analysis is required. In the Appendix, we further discuss sanity tests we executed to try and buttress our second explanation, which we henceforth dub the *hacking* hypothesis.

Despite our difficulty generalizing model stitching, we still see it as an important step forward in our ability to compare representations, since its focus on *functional* similarity makes its results more salient than those from geometric closeness or statistical measures. Unlike existing measures of similarity which tend to look for literal distance between representations, stitching follows a process whereby we define what behaviors should be exhibited by similar representations (i.e. they should be interchangeable up to a degree of flexibility determined by the function class of the stitch) and then devise tasks/experiments that test these behaviors on the representations under question. This approach is far better, because the research community understands tasks much better than the numerical, geometric, or statistical properties of representations. Moreover, it is easier to interpret the resulting similarity measurements in terms of accuracy or other such *functional* quantities, making techniques like stitching more useful, even in practice. We hope to see more representational comparison techniques following the high-level process outlined above in the future.

## Acknowledgments and Disclosure of Funding

Thank you to MIT EECS for funding this project and MIT SuperUROP and the Soljacic Lab for supporting it.

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

# A Appendix

Here we provide additional information that may be helpful to readers, but which was beyond the scope of the main paper. There are four subsections: further details on our testing procedures; results from larger Resnets to support the idea that our surprising findings could generalize; numerical sanity testing to try and confirm or disprove the *hacking* hypothesis; and image-generation to try and visualize the possibility of the *hacking* hypothesis.

**Experimental Details**

We train our stitches for four epochs with momentum 0.9, batch size 256, weight decay 0.01, learning rate 0.01, and a post-warmup cosine learning rate scheduler. We chose our hyperparameters because they were effective for training the Small Resnets between which we stitched. Below is an example of a Small Resnet for clarity on our architecture.

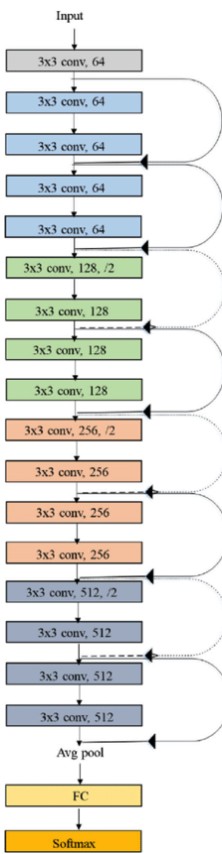

Figure 3: Resnet18 He et al. (2016), equivalent to $R_{2222}$ using our nomenclature.

**Extrapolation to Larger Resnets**

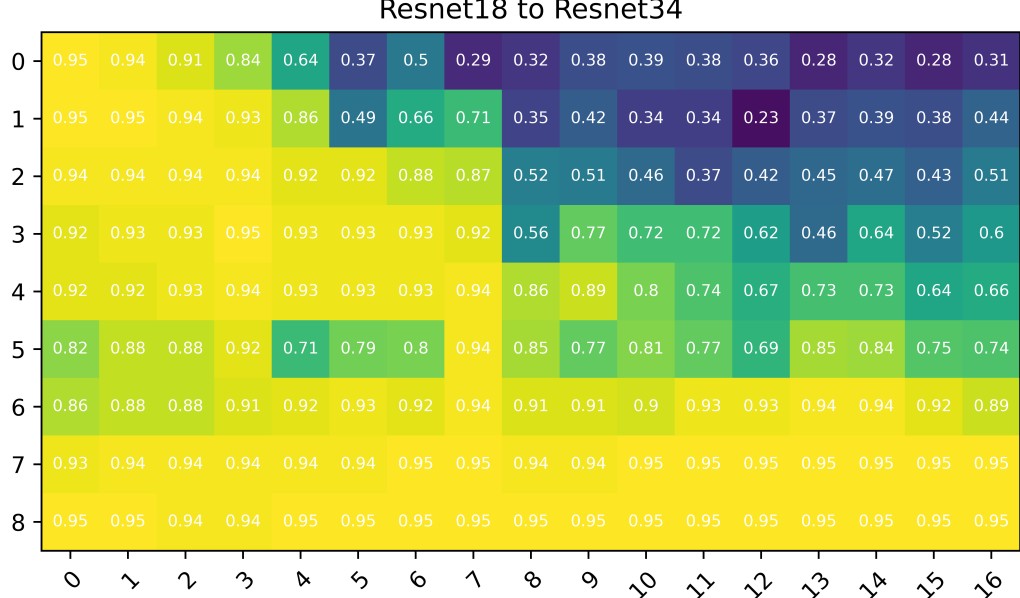

Figure 4: Our results generalize to larger Resnets of a similar type (also on CIFAR-10). We were able to yield the same triangular pattern between Resnet18 and Resnet34 in both directions, suggesting that our results are the function of some general property of the Resnet architecture or of CIFAR-10. One thing that becomes more aparent for Resnet pairs whose lengths are very different, is that it is not whether the reciever layer comes before the sender layer, numerically, that matters, but whether it comes before it proportional to the length of the reciever Resnet. This is perplexing. Visually, in the table this means that the lower left hand triangle of high similarity includes all elements below the diagonal, from the top left cell to the bottom right cell, *regardless of the slope of this diagonal*.

**Numerical Sanity Testing**

When we found that our metric was finding high similarity between distant layers, we decided to sanity test this result using numerical testing. We averaged the mean squared error between three pairs of values: the expected representation with that generated by a vanilla stitch (trained with backpropagation on the CIFAR-10 classification task, as discussed in the body of this paper); the expected representation with that generated by a similarity-trained stitch; and the representation generated by a similarity-trained stitch with that generated by a vanilla stitch. Respectively, we refer to these three pairs as **EV**, **ES**, and **SV**. The mean squared error is over the elements in the representations' tensors. For those mean squared errors, we found the minimum, mean, maximum, and standard deviation over the (cartesian product of the) entire dataset of CIFAR-10 and all the pairs of layers across *all* Small Resnets. We also measured the same statistics for only corresponding layers (i.e. layer one with layer one) to get a baseline similar to what Bansal et. al.'s work may have yielded.

Unlike the vanilla stitch, the similarity-trained stitch was trained to minimize the mean squared error between the expected representation of the reciever (at a layer) and the output of the corresponding stitch. For example, consider the input $x$, sender $A$, reciever $B$, and stitch $S$. Consider the stitched network $C = B_{j<}(S(A_{\leq i}))$, and recall that the expected representation is $B_{\leq j}$ (that is, the computation up to, including, layer $j$ of the reciever). The vanilla stitch would be trained using backpropagation on $C$ with all the weights of $A$ and $B$ frozen (only the

weights in $S$ can change). However, the similarity-trained stitch would be trained on $(S(A_{<i}) - B_{\leq j})^2$. The purpose of the similarity-trained stitch is to get a baseline for what a "low" mean squared error is. We include its task accuracy at the end of this section as a curiosity.

Below we plot our results in two tables. We denote the highest difference with red, that with the second highest difference with yellow, and that with the lowest difference with green. We expect, therefore, to see red in the **EV** and **SV** columns and green in the **ES** column. That is because the similarity-trained stitch should be closer to the expected representation (since the their difference is its loss function) than the vanilla stitch is to either. If the vanilla stitch is learning to hack the reciever then we expect its difference to be larger by many orders of magnitude than the similarity-trained stitch. While we do see that our prediction is typically correct, the difference is not as large nor as decimatingly common as we had hoped, and so we cannot conclude, from these results, that the stitch is likely hacking the reciever.

Table 1: Diagonals

| Minimum | | | Mean | | | Maximum | | | Standard Deviation | | |
|---|---|---|---|---|---|---|---|---|---|---|---|
| EV | ES | SV | EV | ES | SV | EV | ES | SV | EV | ES | SV |
| 2.0e-3 | 2.2e-5 | 1.2e-3 | 4.4e-2 | 1.5e-2 | 2.3e-2 | 2.8e-1 | 1.9e-1 | 1.1e-1 | 6.1e-2 | 3.7e-2 | 3.9e-2 |
| 1.3e-1 | 5.5e-2 | 2.7e-3 | 4.3e+5 | 7.6e+4 | 1.7e+5 | 4.2e+6 | 3.6e+5 | 3.7e+6 | 1.1e+6 | 1.3e+5 | 7.3e+5 |
| 1.5e-2 | 3.5e-5 | 7.9e-3 | 3.5e-1 | 1.1e-3 | 3.2e-1 | 5.3e-1 | 5.9e-3 | 5.3e-1 | 1.6e-1 | 2.0e-3 | 1.6e-1 |
| 1.6e-1 | 1.7e-2 | 1.2e-1 | 1.3e+2 | 4.9e+0 | 1.2e+2 | 1.4e3 | 6.0e+1 | 1.4e+3 | 2.9e+2 | 1.3e+1 | 2.9e+2 |

Table 2: All Stitches

| Minimum | | | Mean | | | Maximum | | | Standard Deviation | | |
|---|---|---|---|---|---|---|---|---|---|---|---|
| EV | ES | SV | EV | ES | SV | EV | ES | SV | EV | ES | SV |
| 1.6e-3 | 5.4e-6 | 7.8e-4 | 4.5e-2 | 1.6e-2 | 2.0e-2 | 3.6e+0 | 1.3e+0 | 5.9e-1 | 1.4e-1 | 5.9e-2 | 3.3e-2 |
| 1.6e-4 | 1.7e-7 | 1.5e-4 | 2.6e+5 | 1.1e+5 | 6.3e+4 | 1.8e+7 | 9.2e+6 | 5.8e+6 | 1.0e+6 | 4.3e+5 | 3.4e+5 |
| 1.4e-2 | 6.3e-6 | 7.9e-3 | 2.2e+0 | 2.1e-3 | 2.2e+0 | 2.3e+2 | 1.9e-2 | 2.3e+2 | 1.2e+1 | 4.3e-3 | 1.2e+1 |
| 1.3e-1 | 1.6e-2 | 9.4e-2 | 7.3e+4 | 4.2e+4 | 2.1e+4 | 1.2e+7 | 6.9e+6 | 4.8e+6 | 5.9e+5 | 3.4e+5 | 2.1e+5 |

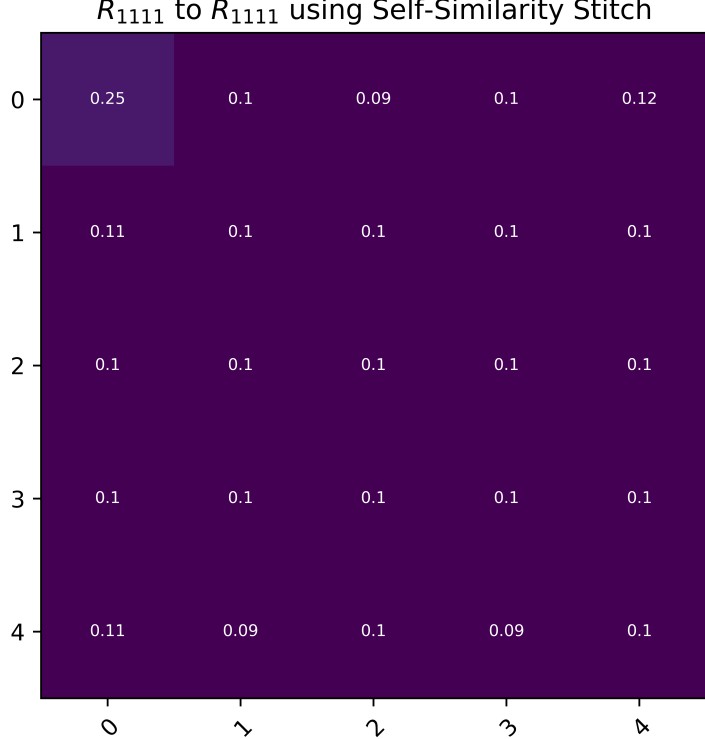

Figure 5: We are unable to yield high task accuracy for stitched networks using similarity-trained stitches. This makes sense, since having no task information, it does not know what subspaces to prioritize. Most likely, only a few subspaces and/or weights truly matter for task accuracy, per the Lottery-Ticket hypothesis Frankle & Carbin (2018). Thus, not knowing which those are, the similarity-trained stitch cannot yield high task accuracy even when it is trained for thirty epochs to the vanilla stitch's four (the latter yielding a very high similarity near 90% for the lower left-side triangle of the corresponding similarity matrix, per the previous figures in this paper).

**Image Generation**

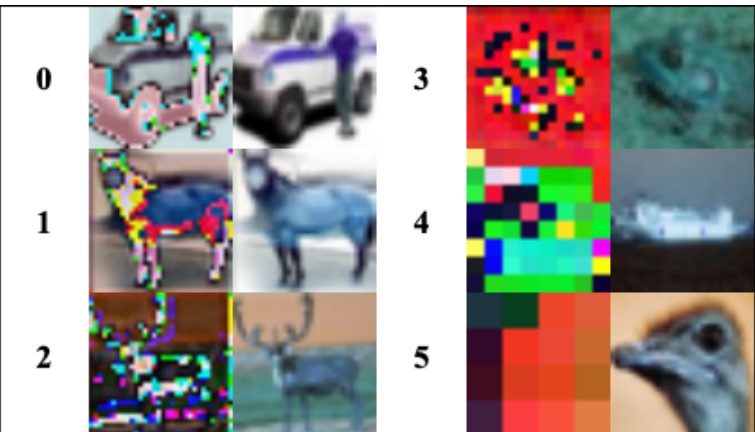

Figure 6: Our results from generating images using stitches. We stitched the output of intermediate layers (numbered on the left from zero through five to signify the number of blocks coming before it—zero corresponding to the output of the first convolution, for example) into the very first layer, thereby generating images. We hoped to understand whether the stitches were able to hack the reciever. We did, however, not find a discernable pattern other than the loss of granularity as the layers progressed. On the left side of each pair is the stitch-generated image, whereas on the right side is the actual image.

