# OpenReview forum: "Model Stitching: Looking For Functional Similarity Between Representations"
_NeurIPS.cc/2022/Workshop/SVRHM — SVRHM Poster_

### Official Review · Reviewer_GsPq · 2022-10-12
**Interesting findings worth publishing, but more discussion of what they mean and how they relate to other studies would be useful.**

**Rating:** 6
**Confidence:** 3

**Review:**


Interesting finding suggesting that Model stitching can provide misleading estimates of the similarity of representations between models.  I think this issue is important, and the studies seem well done.  But the authors could do better at explaining how their findings relate to other work.
1)	 More discussion about why the findings appear at odds with Bansal et al.  Why the different results?
2)	Bit more discussion of how high similarity between representations have been interpreted.
3)	There is some other work highlighting how RSA and predictivity can produce misleading estimates of similarity that might be cited.  For example:

Dujmović, M., Bowers, J., Adolfi, F., & Malhotra, G. (2022). The pitfalls of measuring representational similarity using representational similarity analysis. bioRxiv.
Schaeffer, R., Khona, M., & Fiete, I. (2022). No free lunch from deep learning in neuroscience: A case study through models of the entorhinal-hippocampal circuit. bioRxiv.

Minor point: The formatting of references needs improvement.

---

### Official Review · Reviewer_wrtS · 2022-10-14
**An extensive and well presented set of results**

**Rating:** 7
**Confidence:** 3

**Review:**

This paper analyses model stitching (the act of training a translation layer which maps from a representation in a sender network to an alternate representation in a receiver) for a wide range of ResNet variants, with a view to analysisng it's efficacy as a measure of functional similarity. The authors find that stichting from later layers in the sender to earlier layers in the receiver can reach high accuracy and not vice versa.

The experiments conducted in the paper are extensive and the results are clearly presented. Furthermore, assessing functional similarity is an important problem and this work makes a clear contribution to the discourse.

Beyond the rigourous experimentation, the paper avoids making any in depth assessment of exactly how the results could come about. This would be an appeciated addition since I'm not sure the results are as surprising as they seem at the surface. One view of deep classification networks is that they discard information with depth until only the class information remains. Similary, as depth increases (or rather, distance from the output decreases) it becomes increasingly easy to linearly transform from the representation to the class. So, for a simple network to map from one representation to another (given that it only needs to retain class information in it's output to succeed) should become easier with depth as the authors observe here.

In the limit of that argument, consider a single linear layer which maps the output of one network to the input of another. Such a network would only need to learn an adversarial example for each class in the weight space to solve the stitching perfectly (and in a way that would be much harder if taking input from earlier layers). The authors touch on a similar argument in their conclusions and breifly in an appendix, but I think that there is scope for future presentations of this work to delve much deeper into that idea and be much bolder in the presentation of their hypotheses.

Overall, this paper is clearly of value to the SVRHM community and so I vote to accept. That said, I would be very interested to read a much more detailed assessment of the authors theories on their findings and what they mean for the validity of such approaches to determining functional similarity.

---

### Official Review · Reviewer_rHH1 · 2022-10-14

**Rating:** 6
**Confidence:** 4

**Review:**

This paper shows that for ResNets trained on CIFAR-10, a sender network layer can be stitched to any layer that precedes that layer in the receiver network to achieve surprisingly high accuracy. The paper hypothesizes that this result implies that stitches are not necessarily matching expected representations, but rather finding different ones that achieve similar results, and therefore may not be a measure of representational similarity.

The result that a sender representation from layer i can perform well stitched into layer j in the receiver (where j < i) but a representation from sender layer j cannot be stitched into receiver layer i in the receiver is indeed interesting. However, it is not clear that this result implies that stitching is not measuring representation similarity. One alternative explanation for this result would be that this experiment reparametrizes the stitching function into the composition of a learnable layer S and non-learnable receiver network layers between j and i. This is especially plausible for residual architectures that may contain collapsing paths or other “fixed point” dynamics [1]. For instance, when stitching a later layer from a trained sender to a random receiver, the function achieves high accuracy even for early receiver layers. It would be useful to see a control for this explanation. Alternatively, consistent results on non-residual architectures would increase the strength of the hypothesis.

Minor comments:
- When stitching from resnet18 to resnet34, high accuracy is achieved even when stitching to much later layers of the network. Is there some explanation for this?
- The paper states that networks are trained for each type of small ResNet, but results are only reported for R_1111 and R_1122. Are these results representative of other stitches? It would be good to get an aggregated view across small ResNet sizes.

[1] Invertible Residual Networks. Behrmann et al.